# Influence of Preoperative Diagnosis of Nutritional Disorders on Short-Term Outcomes After Hip Arthroplasty: A Cohort Study of Older Adults

**DOI:** 10.3390/nu17142319

**Published:** 2025-07-14

**Authors:** Matteo Briguglio, Marialetizia Latella, Paolo Sirtori, Laura Mangiavini, Paola De Luca, Manuela Geroldi, Elena De Vecchi, Giovanni Lombardi, Stefano Petrillo, Thomas W. Wainwright, Giuseppe M. Peretti, Giuseppe Banfi

**Affiliations:** 1IRCCS Ospedale Galeazzi—Sant’Ambrogio, Laboratory of Nutritional Sciences, 20157 Milan, Italy; 2IRCCS Ospedale Galeazzi—Sant’Ambrogio, EUORR Unit, 20157 Milan, Italy; 3Department of Biomedical Sciences for Health, University “La Statale”, 20122 Milan, Italy; 4IRCCS Ospedale Galeazzi—Sant’Ambrogio, Laboratory of Biotechnology Applied to Orthopedics, 20157 Milan, Italy; 5IRCCS Ospedale Galeazzi—Sant’Ambrogio, Laboratory of Clinical Chemistry and Microbiology, 20157 Milan, Italy; 6IRCCS Ospedale Galeazzi—Sant’Ambrogio, Laboratory of Experimental Biochemistry and Molecular Biology, 20157 Milan, Italy; 7Department of Athletics, Strength and Conditioning, Poznań University of Physical Education, 61-871 Poznań, Poland; 8IRCCS Ospedale Galeazzi—Sant’Ambrogio, Prosthetic Surgery Centre, 20157 Milan, Italy; 9Orthopaedic Research Institute, Bournemouth University, Bournemouth BH12 5BB, UK; 10NHS Foundation Trust, University Hospitals Dorset, Bournemouth BH7 7DW, UK; 11Rehabilitation Department, Lanzhou University, Lanzhou 730000, China; 12IRCCS Ospedale Galeazzi—Sant’Ambrogio, Scientific Direction, 20157 Milan, Italy; 13Faculty of Medicine and Surgery, Vita-Salute San Raffaele University, 20132 Milan, Italy

**Keywords:** orthopaedic procedures, total hip replacement, prehabilitation, malnutrition, sarcopenia, enhanced recovery after surgery, postoperative complication, quality improvement, patient care

## Abstract

**Background**: Nutritional disorders may affect short-term recovery after major orthopaedic surgery, but evidence is lacking. This study assessed whether and how different nutritional disorders diagnosed at admission could influence early recovery after hip replacement. **Methods**: A prospective analytical study was designed to include 60 patients scheduled for elective primary hip replacement and assess their nutritional status to diagnose 5 malnutrition phenotypes: undernutrition, sarcopenia, obesity, sarcopenic obesity, and sarcopenic undernutrition. Outcome measures were 24 h change in neutrophils, 72 h change in haemoglobin, and 10-day gait speed regain. **Results**: Haemoglobin reached the nadir at day 2–3 and partially recovered by day 10 in all patients, with sarcopenia and undernutrition being the strongest predictors of the postoperative drop (−2.37 g∙dL^−1^ and −0.80 g∙dL^−1^, *p* < 0.05). Neutrophils peaked immediately after surgery and returned to baseline levels at discharge, with sarcopenic undernutrition displaying a blunted response after surgery (−16.20%, *p* < 0.01). Undernutrition was found to be the most influential preoperative variable on gait speed recovery, but with a marginal effect. None of the patients covered the reference energy and protein needs through diet in the 10 postoperative days. **Conclusions**: In this cohort, nutritional disorders with reduced body function and reserves (sarcopenia and undernutrition) grounded a greater vulnerability to surgery in terms of early stress response and short-term recovery. This calls for both advanced planning of nutritional prehabilitation strategies for these conditions and adequate postoperative nutritional support.

## 1. Introduction

Nutritional disorders include alterations in body composition to the detriment of lean mass and functioning (sarcopenia), pathological underweight (undernutrition), and excessive weight with extra fat mass (obesity) [1]. These disorders can occur regardless of the patient’s age and health conditions, although they are more easily found in older individuals and those with polymorbidity. Common aetiological factors are either a reduced appetite that pushes the patient to hypophagia, causing the failure to meet nutrient requirements, or a hyperphagia that causes an excess of, most often, calorie intake [2]. Algorithms to classify nutritional disorders are relatively recent [3,4,5], with the diagnosis that requires time, specialised equipment, and expertise. To date, studies investigating the predictive potential of nutritional disorders on postoperative outcomes in the field of major orthopaedic surgery and, specifically in joint replacement, are lacking. Malnutrition has historically been identified both in orthopaedic research and clinical practice based on surrogate parameters, such as body mass index (BMI), circulating levels of some analytes of nutritional interest, like albumin, transferrin, or total lymphocyte counts, and composite equations [6,7]. There is plenty of evidence associating low circulating levels of these surrogate parameters to deleterious consequences in major orthopaedic surgery, ranging from delayed recovery to higher risk of infections of the joint endoprosthesis or surgical site [8,9]. However, these metrics come with limitations [10] and do not conform to the recognised diagnostic criteria. The most recent studies have begun to use assessment methods that are considered more appropriate for malnutrition and other nutritional disorders like sarcopenia, such as the quantities of fat and lean mass, strength, and function [11,12]. Still, the application of the diagnostic criteria in full should remain the approach of choice in prospective studies that aim both to diagnose nutritional disorders and to investigate if there is any influence on recovery. This is critical to understand which type of nutritional prehabilitation can be of use, thus leading to improvements in the quality of perioperative care in major orthopaedic surgery for patients with nutritional disorders.

In this analytical investigation, we studied different recovery trends up to ten days after hip replacement surgery in relation to the presence of a diagnosed nutritional disorder at hospital admission.

## 2. Materials and Methods

### 2.1. Study Design and Setting

The study was designed as a prospective analytical study to be conducted in our highly specialised hospital in orthopaedic surgery (IRCCS Ospedale Galeazzi—Sant’Ambrogio, Milan, Italy). The study cohort included older patients admitted for elective hip replacement and for whom discharge was expected within 72 h of the operation according to the standard protocol of surgical and anaesthetic practices. Briefly, blood management included the cessation of antiplatelets in primary or titration in secondary prophylaxis, cessation or bridging of anticoagulants depending on the patient’s pharmacotherapy and co-existing diseases, a maximum dose of 1 g of tranexamic acid intravenous before surgery and intra-articular before wound closure, intraoperative transfusion at discretion of the anaesthetist, postoperative oral iron therapy at discretion of the surgeon, and postoperative transfusions if there was concomitant presence of excessive haemoglobin drop, symptomatic anaemia (e.g., hypotension, tachycardia, and asthenia), and electrocardiogram abnormalities.

Patients underwent a nutritional assessment and diagnosis before the operation, a perioperative monitoring of routinely collected blood parameters, a daily record of food intakes, and a postoperative re-assessment of physical function. Specifically, study evaluations were planned to be performed at hospital admission, discharge, and at the follow-up visit after 10 days.

### 2.2. Participants

Patients referring to a single surgical team (EUORR Unit) were screened and recruited at the admission visit, which could correspond to the day before or the same day of surgery. Male and female patients between 60 and 85 years of age scheduled for primary total hip replacement were eligible. Patients with a major neurological or psychiatric condition, advanced heart or kidney disease, cancer, or those unable to adhere to the study evaluations were excluded. The follow-up assessment at discharge was performed at the patient’s bedside at day 3, while the 10-day evaluation was conducted in a dedicated outpatient clinic within the hospital.

### 2.3. Variables and Methods

Nutritional disorders at admission were assumed to be exposures/predictors. The time points of analysis encompassed the time before the operation (preop), immediately after (op), the second evaluation after (postop), one day after (day 1), two days after (day 2), discharge (day 3), and the check-up visit approximately 10 days after the operation (day 10). Three endpoint variables were chosen for the predictive analyses on postoperative recovery: the 72 h haemoglobin change (nadir), the 24 h neutrophil change, and the 10-day gait speed change. Baseline effect modifiers or potential confounders were anticipated to be sex, age, and co-existing illnesses. Other than the predictive analyses conducted on the three endpoint variables, in this article we report other postoperative trends that could give a more complete picture of the haematological (red blood cells, haemoglobin, haematocrit, mean corpuscular volume, mean corpuscular haemoglobin, mean corpuscular haemoglobin concentration, iron, and ferritin) and inflammatory (neutrophils, lymphocytes, neutrophil-to-lymphocyte ratio, monocytes, eosinophils, basophils, sclerostin, and dickkopf-related protein 1) response, recovery of physical function (weight, handgrip strength, phase angle and body cell mass by bioimpedance analysis, 10 m walk test, timed up and go test, and Barthel index), and food consumption (daily protein–energy intakes).

Diagnostic criteria of nutritional disorders, equipment used, and methods of assessment have been described elsewhere [13,14]. Briefly, undernutrition required the co-presence of unintentional weight loss, low age-adjusted BMI, or low sex-adjusted muscle mass, severe gastrointestinal condition, disease burden, or inflammation, and low phase angle. Sarcopenia required the co-presence of low sex-adjusted handgrip strength, low sex-adjusted muscle mass, and poor performance through the 10 m walk test or timed up and go test. Obesity required the co-presence of a high BMI, low sex-adjusted handgrip strength, excess of sex-adjusted fat mass, and reduced sex-adjusted muscle mass. Regarding the patients’ diet at home, participants completed a paper-based quantitative food diary (weighting of food required) in the first 10 days after the intervention, returning it at the final visit.

### 2.4. Statistics

The study size was calculated considering a haemoglobin change of −3.3 g/dL within 72 h from surgery as the primary endpoint, which led to a sample size calculation of 60 patients considering a 10% loss prediction. No resampling technique was planned to retrieve potential dropout patients. Missing data in repeated outcome measured were imputed with the last observation carried forward method. The Friedman test was used to check if at least one time point of the outcome measures (dependent variables) was significantly different in well-nourished patients (reference) after surgery (independent variable was “time”). The Wilcoxon signed-rank test was conducted to test the same hypothesis for sclerostin and dickkopf-related protein 1. Generalised estimating equations (GEEs) estimated the independent main effects of time without interactions on postoperative values of haemoglobin and neutrophils across seven time points (preop, op, postop, day 1, day 2, day 3, and day 10) and of gait speed at preop, day 3, and day 10. The GEE two-way model (reference category = well nutrition) adjusted for sex, age, comorbidity index, and nutritional disorder was used to explore the interaction effects on haemoglobin, neutrophils, and gait speed trends over time. Hierarchical multiple regression explored how much additional variance in 72 h haemoglobin change, 24 h neutrophil change, and 10-day gait speed change could be explained by the baseline nutritional disorder beyond the effects of age, sex, and comorbidity index. Elastic net regression was used to select via 10-fold cross-validation the best predictors of 72 h haemoglobin, 24 h neutrophil, and 10-day gait speed change among sex, age, comorbidity index, and nutritional disorder. No analysis was planned on the other collected parameters and recovery measures to avoid the risk of multiplicity and the likelihood of finding spurious associations. Moreover, different variables would have shown overlapping trends (e.g., haemoglobin and haematocrit), possibly conveying redundant information or adding an analytical complexity without proportionally enhancing the interpretation. Statistics and machine learning were performed using the R programming language (version 2024.04.2+764) by means of dplyr (version 1.1.4), tidyr (version 1.3.1), rstatix (version 0.7.2), geepack (version 1.3.12), glmnet (version 4.1-8), and broom (version 1.0.7) packages. This article follows the guidelines for reporting observational cohort studies (STROBE statement).

## 3. Results

The cohort characteristics at baseline have been previously described in two preliminary reports [13,14]. We recorded no deviations from the hospital standard of care for what concerned surgical or anaesthetic practices and blood management strategies, such as emergency transfusions or complementary oral iron therapy. The presence or absence of a nutritional disorder could be assessed in 48 out of 60 participants (3 dropped for postponed surgery and 9 with unreliable data for diagnosis). Twenty-six patients were well nourished, nine had pure obesity, five had pure undernutrition, two had pure sarcopenia, four had sarcopenic undernutrition, and two had sarcopenic obesity. Missing values in the imputed dataset were 3.6%. Baseline data of the 48 patients are summarised in Table 1, Table 2, Table 3 and Table 4.

Among the 26 well-nourished individuals, the Friedman test showed no significant difference after surgery between the mean ranks of body weight (χ^2^(2) = 4.29, *p* = 0.117), handgrip strength (χ^2^(2) = 1.91, *p* = 0.385), phase angle (χ^2^(2) = 4.47, *p* = 0.107), or body cell mass (χ^2^(2) = 3.27, *p* = 0.195). The Wilcoxon signed-rank test showed no statistically significant median change in both sclerostin (V = 40, *p* = 0.277) and dickkopf-related protein 1 (V = 63, *p* = 0.890) levels from before to after surgery. A significant change was found in postoperative gait speed (χ^2^(2) = 23.57, *p* < 0.0001), timed up and go (χ^2^(2) = 27.42, *p* < 0.0001), Barthel index (χ^2^(2) = 20.51, *p* < 0.0001), protein intakes (χ^2^(9) = 37.38, *p* < 0.0001), calorie intakes (χ^2^(9) = 54.59, *p* < 0.0001), neutrophils (χ^2^(6) = 62.76, *p* < 0.0001), lymphocytes (χ^2^(6) = 62.81, *p* < 0.0001), monocytes (χ^2^(6) = 60.72, *p* < 0.0001), eosinophils (χ^2^(6) = 62.66, *p* < 0.0001), basophils (χ^2^(6) = 59.15, *p* < 0.0001), neutrophil-to-lymphocyte ratio (χ^2^(6) = 63.26, *p* < 0.0001), red blood cells (χ^2^(6) = 116.04, *p* < 0.0001), haemoglobin (χ^2^(6) = 114.47, *p* < 0.0001), haematocrit (χ^2^(6) = 111.65, *p* < 0.0001), mean corpuscular volume (χ^2^(6) = 14.20, *p* = 0.027), mean corpuscular haemoglobin (χ^2^(6) = 14.58, *p* = 0.024), mean corpuscular haemoglobin concentration (χ^2^(6) = 8.17, *p* = 0.226), iron (χ^2^(2) = 38.86, *p* < 0.0001), and ferritin (χ^2^(2) = 34.93, *p* < 0.0001).

### 3.1. Red Blood Cell and Iron Status Indices

GEE analysis of main effects showed that the haemoglobin concentration significantly decreased in all participants (*p* < 0.001) regardless of baseline factors. Values reached the minimum at day 2 (β = −3.19 g∙dL^−1^, SE = 0.15, 95% CI [−3.49, −2.90], *p* < 0.001) and remained low until discharge at day 3 (β = −3.11 g∙dL^−1^, SE = 0.15, 95% CI [−3.39, −2.83], *p* < 0.001). After ten days, there was a slight increase (β = −2.27 g∙dL^−1^, SE = 0.14, 95% CI [−2.55, −1.99], *p* < 0.001). See Figure 1b for details. Adjusting for sex, age, comorbidity index, and nutritional disorder (interaction model), postoperative levels of haemoglobin significantly declined at most postoperative time points compared to baseline (op, *p* = 0.789; postop, β = −2.71 g∙dL^−1^, SE = 1.26, 95% CI [−5.18, −0.24], *p* = 0.032; day 1, β = −2.94 g∙dL^−1^, SE = 1.42, 95% CI [−5.72, −0.16], *p* = 0.038; day 2, β = −3.50 g∙dL^−1^, SE = 1.51, 95% CI [−6.46, −0.53], *p* = 0.021; day 3, β = −4.67 g∙dL^−1^, SE = 1.70, 95% CI [−8.01, −1.32], *p* = 0.006; day 10, *p* = 0.543).

An interaction effect on postoperative haemoglobin trends was found for comorbidity index and nutritional disorder × time. No association was found with sex or age (*p* > 0.05). Specifically, a higher comorbidity index was associated with greater haemoglobin decline at day 3 (β = −0.32 g∙dL^−1^, SE = 0.12, 95% CI [−0.52, −0.09], *p* = 0.006). Patients with sarcopenia showed a greater drop in haemoglobin over time compared to well-nourished individuals, with major effects observable at day 2 (β = −1.91 g∙dL^−1^, SE = 0.44, 95% CI [−2.77, −1.04], *p* < 0.001) and day 3 (β = −2.37 g∙dL^−1^, SE = 0.33, 95% CI [−3.03, −1.70], *p* < 0.001). Undernourished patients experienced a greater haemoglobin drop than well-nourished patients at postop (β = −0.80 g∙dL^−1^, SE = 0.40, 95% CI [−1.58, −0.02], *p* = 0.048). The sarcopenic undernutrition phenotype also had a significant change at day 10 (β = 1.12 g∙dL^−1^, SE = 0.43, 95% CI [0.27, 1.96], *p* = 0.010). No significant interactions were found between time and obesity or sarcopenic obesity (*p* > 0.05).

The hierarchical multiple regression model with age, sex, and comorbidity index explained 6.72% of the variance in 72 h haemoglobin change (R^2^ = 0.0672, *p* = 0.377; β(age) = 0.0205, SE = 0.0256, *p* = 0.428; β(sex) = −0.2803, SE = 0.3062, *p* = 0.365; β(comorbidity) = −0.2384, SE = 0.1399, *p* = 0.096). Adding the nutritional disorder factor increased the variance explained to 30.3%, with the overall model being significant (*p* = 0.038). The sarcopenia variable (β = −2.37 g∙dL^−1^, SE = 0.73, 95% CI [−3.84, −0.98], *p* = 0.0024) was significantly associated with a greater haemoglobin reduction at 72 h. The elastic net model selected sarcopenia (β = −0.917 g∙dL^−1^) and undernutrition (β = −0.257 g∙dL^−1^) as the only predictors of 72 h haemoglobin change, with age, sex, comorbidity index, and other malnutrition phenotypes having their coefficients shrunk to zero.

### 3.2. Immune and Inflammation Parameters

Compared to baseline levels, neutrophils had higher levels at the first (β = 13.48%, SE = 1.50, 95% CI [10.54, 16.42], *p* < 0.001) and second blood tests after surgery (β = 11.68%, SE = 1.31, 95% CI [9.11, 14.26], *p* < 0.001), then significantly declined at day 1 (β = 5.91%, SE = 1.18, 95% CI [3.59, 8.23], *p* < 0.001) and matched preoperative levels at day 3 (*p* = 0.258; see Figure 2a).

After adjusting for baseline co-factors, neutrophils significantly changed only at day 2 (op, β = 19.37%, SE = 13.89, 95% CI [−7.82, 46.55], *p* = 0.16; postop, β = 8.04%, SE = 15.63, 95% CI [−22.58, 38.66], *p* = 0.61; day 1, β = −18.21%, SE = 12.39, 95% [−42.49, 6.08], *p* = 0.14; day 2, β = −31.46%, SE = 12.77, 95% CI [−56.51, −6.41], *p* = 0.014; day 3, β = −23.75%, SE = 14.09, 95% CI [−51.35, +3.85], *p* = 0.092; day 10, β = −19.02%, SE = 10.47, 95% CI [−39.54, 1.51], *p* = 0.069). Sex did not influence the postoperative trends of neutrophils nor the comorbidity index (*p* > 0.05). Conversely, older participants exhibited a prolonged neutrophil raise at day 2 (β = 0.51, SE = 0.19, 95% CI [0.14, 0.88], *p* = 0.007), day 3 (β = 0.42, SE = 0.22, 95% CI [0.00, 0.85], *p* = 0.051), and day 10 (β = 0.30, SE = 0.15, 95% CI [0.01, 0.59], *p* = 0.046). The interaction model of GEE showed that the change in neutrophil levels over time differed in patients suffering from sarcopenia (op, β = 15.93%, SE = 3.05, 95% CI [10.95, 20.91], *p* < 0.0001; day 10, β = 8.96%, SE = 2.00, 95% CI [5.05, 12.86], *p* < 0.0001) and sarcopenic undernutrition (postop, β = −16.20%, SE = 5.84, 95% CI [−27.64, −4.76], *p* = 0.0055; day 1, β = −14.78%, SE = 4.02, 95% CI [−22.66, −6.90], *p* < 0.0001; day 2, β = −14.35%, SE = 4.78, 95% CI [−23.73, −4.97], *p* = 0.0027; day 3, β = −12.75%, SE = 4.37, 95% CI [−21.31, −4.19], *p* = 0.0035; day 10, β = −9.97%, SE = 4.01, 95% CI [−17.83, −2.11], *p* = 0.0129) compared to well-nourished subjects. Non-significant interaction effects were found for obesity, sarcopenic obesity, or undernutrition (*p* > 0.05).

The hierarchical multiple regression model explained 3.42% of the variance in 24 h neutrophil change before adjustment with nutritional disorder (R^2^ = 0.0342, *p* = 0.671; β(age) = −0.0646, SE = 0.2332, *p* = 0.780; β(sex) = −0.2712, SE = 2.7907, *p* = 0.920; β(comorbidity) = −1.2602, SE = 1.2753, *p* = 0.330) and 25.7% after adjustment (*p* = 0.132). Sarcopenic undernutrition (β = −16.20%, SE = 4.86, 95% CI [−26.07, −6.33], *p* = 0.0019) was significantly associated with a more modest increase in neutrophils at 24 h. Among sex, age, comorbidity index, and nutritional disorder, the elastic net model selected sarcopenic undernutrition (β = −9.59%) as the only predictor of 24 h neutrophil change.

### 3.3. Physical Function

Gait speed (Figure 3e) significantly reduced at day 3 (−0.28 m∙s^−1^, SE = 0.04, 95% CI [−0.35, −0.21], *p* < 0.001), with a recovery at day 10 that brought the performance scores not different from those at baseline (0.02 m∙s^−1^, SE = 0.05, 95% CI [−0.07, 0.10], *p* = 0.684). The GEE model with interaction effects confirmed a main effect of time at day 3 (−1.09 m∙s^−1^, SE = 0.36, 95% CI [−1.79, −0.38], *p* = 0.002) and a recovery at day 10 (−0.12 m∙s^−1^, SE = 0.47, 95% CI [−1.04, 0.80], *p* = 0.795). Neither sex nor the comorbidity index (*p* > 0.05) influenced the postoperative trend of gait speed. In contrast, there was an interaction effect with age at day 3 (0.011 m∙s^−1^, SE = 0.005, 95% CI [0.001, 0.021], *p* = 0.032) but not at day 10 (*p* = 0.520). None of the nutritional disorder categories significantly interacted with time.

The hierarchical multiple regression model with age, sex, and comorbidity index explained 2.81% of the variance in 10-day gait speed change (R^2^ = 0.0281, *p* = 0.744; β(age) = 0.00283, SE = 0.00812, *p* = 0.730; β(sex) = −0.03690, SE = 0.09933, *p* = 0.710; β(comorbidity) = −0.05042, SE = 0.04553, *p* = 0.270). Adding the nutritional disorder factor increased the variance explained to 8.71% (*p* = 0.881), with no malnutrition phenotype showing a statistically significant association with 10-day gait speed change. Among sex, age, comorbidity index, and nutritional disorder, the elastic net model shrunk all the coefficients to zero except the one of undernutrition, which was, however, a small effect (β = −4.57^−17^ m∙s^−1^).

### 3.4. Postoperative Dietary Intakes

All 48 patients assessed for the presence of a nutritional disorder completed and returned the paper food diary at the follow-up visit. In Figure 3, the dietary intakes, as estimated from the calculations, are shown. Up until day 3, the patients followed the hospital diet, while from day 4 to day 10 the data reported are those of home consumption. The reference intervals for optimal intakes per kilogram of patient’s body weight were set between 1.3 and 1.5 g for proteins (Figure 3g) and between 25 and 30 for calories (Figure 3i). Significant increases in energy intake were observed after surgery, particularly at day 2 (*p* = 0.0015), day 9 (*p* = 0.0036), and day 10 (*p* = 0.0043). Similarly, protein intake significantly increased at day 2 (*p* = 0.0001), day 8 (*p* = 0.032), day 9 (*p* = 0.0019), and day 10 (*p* = 0.0021). Males had significantly higher energy intakes at day 4 (*p* = 0.044), day 9 (*p* = 0.003), and day 10 (*p* = 0.007), along with significantly higher protein intakes from day 3 to day 10 (*p* < 0.05). Older individuals reduced both energy and protein intakes at day 2, day 9, and day 10 (*p* < 0.05). Higher comorbidity index increased both energy and protein intakes at day 2, day 9, and day 10 (*p* < 0.05). These findings were mirrored by a moderate and statistically significant positive association between age and comorbidity index (ρ = 0.439, *p* = 0.002) found by the Spearman’s rank correlation test. Compared to well nutrition, we found a significant association of both sarcopenia and sarcopenic obesity with reduced energy and protein intakes over time, particularly at days 2, 4, 5, 7, 9, and 10. Sarcopenic undernutrition was associated with increased energy and protein intakes at day 2, day 5, and day 9. Undernutrition did not significantly influence protein intake but had some positive effects on energy intake at day 2, day 5, and day 9. No significant effects were seen for obesity.

## 4. Discussion

The objective of this study was to explore the influence of five different nutritional disorders (undernutrition, sarcopenia, obesity, sarcopenic undernutrition, and sarcopenic obesity) on haemoglobin drop, neutrophils’ surge, and gait speed recovery within 10 days after elective hip replacement surgery. These recovery measures are considered clinically meaningful in orthopaedic practice and can serve as robust proxies for the acute-phase response and regain of autonomy post-surgery.

A total of 22 patients, up to 45.8% of the study sample, were diagnosed with a nutritional disorder. Specifically, the sarcopenia phenotype was diagnosed in 16.7%, undernutrition in 18.8%, and obesity in 22.9%. Among these, the multivariate and elastic net analyses found that among various patient factors, nutritional disorders, particularly sarcopenia and undernutrition, were consistently the most influential on early postoperative recovery after hip arthroplasty. Specifically, patients with sarcopenia and undernutrition were those experiencing the worst 72 h decline of haemoglobin (discharge), with the sarcopenia phenotype explaining 30.3% of haemoglobin change variability. These findings suggest that sarcopenia may cause a compromised erythropoietic response post-surgery, possibly due to underlying altered iron metabolism. Moreover, patients with co-existing sarcopenia and undernutrition had a blunted postoperative neutrophil peak, which is consistent with a basal immune dysfunction, and the undernutrition phenotype was also the most significant influencer of the magnitude of the recovery of walking autonomy at follow-up. These key results emphasise that beyond patients’ age, comorbidity burden, and phenotypes of overnutrition (i.e., obesity), a nutritional disorder with underlying reduced body function and reserves can play a critical role in determining the early physiological response after a major orthopaedic surgery. Still, compared to age, that was nevertheless associated in our cohort with prolonged neutrophil elevation (i.e., poorer reactivity or a delayed resolution of the inflammatory response) and greater gait impairment at discharge, malnutrition has the advantage of being a modifiable determinant of health.

Giving the novelty of diagnostic criteria for the various nutritional disorders, few studies are available. A cross-sectional survey has recently investigated the prevalence of sarcopenia and obesity in total joint replacement patients in the United Stated based on body composition analysis, confirming that prevalence of sarcopenia is around 15%, with approximately 65% of patients having obesity [11]. While sarcopenia occurrence is more in line with our results, the second finding is likely a reflection of the different prevalence of obesity in the general Italian and American populations. Another paper recently reported the findings from database research conducted in Spain that mainly aimed at investigating the association of undernutrition and several outcome measures after total hip replacement [15]. Compared to what we observed, the authors found a higher prevalence of both undernutrition, around 38.4%, and obesity, which was around 40%. These data can be due to the fact that our sample consisted of subjects who were on average four years younger, thus missing the burden of ageing on body composition. The Spanish authors also reported that undernutrition was significantly more frequent in the group of patients with postoperative complications, encompassing anaemia requiring transfusions, acute kidney injury, and surgical site infection. This is more in line with our observations, although our hospital standard of care included more stringent surgical and anaesthetic practices for the patients selected in this study, which may have excluded more complex patients for whom the complication rates may have been evident. In contrast to other studies, a diagnosis of obesity was not associated with negative effects on short-term recovery measures in our sample. This may be due to the fact that our patients with obesity had a mean BMI of 33.5 kg∙m^−2^, assigning them to the class 1 obesity. Studies that reported correlations between BMI-derived obesity and increased complications after joint replacement surgeries predominantly focused on patients with class 3 obesity or higher [10,16]. Regarding postoperative dietary intakes, they were far from the optimal reference per kilo of body weight of 25–30 kcal and 1.3–1.5 g of proteins, respectively. The failure of older orthopaedic patients to meet the nutritional needs is a long-known issue that has already been attributed to several aetiologies, including the ageing-derived early satiety, surgery-derived immobility, and polypharmacotherapy [2]. Since it is not interpretable from our results, it remains to be defined whether the different food intakes by sex, age, comorbidity burden, and nutritional diagnosis could be partly influenced by different eating assistance in the ward. Nevertheless, the presence of a nutritional disorder can make things worse, suggesting that malnourished patients may exhibit distinct eating behaviours in response to surgery. Patient education aimed at achieving postoperative dietary needs may be helpful if already planned preoperatively [17].

The additional trends of collected parameters have not been subjected to statistical and predictive interpretations but are presented in comprehensive line charts, offering transparency without diverting from the central informative focus. Overall, many of these changed postoperatively even in the 26 (54.7%) well-nourished patients. This suggests that variations of blood analytes may be attributable to the physiological response to surgery and are not necessarily to be considered harmful. Similarly, well-nourished patients did not experience a significant change in body weight or composition, unlike malnourished patients (Figure 3a,b). The question then arises as to how to distinguish the physiological consequences to the operation from an aberrant response. In our study, we observed that regardless of the baseline nutritional status, all patients experienced a haematological depletion of reserves (e.g., haemoglobin and iron), a leukocytic response that showed a concomitant surge of inflammatory cells (e.g., neutrophils and monocytes) and decline of nutrition-related cells (e.g., lymphocytes), and a reduction in functional status after surgery. It is plausible to think that the solution lies in paying attention both to the patient’s characteristics at admission and to the magnitude and kinetics of the postoperative trends of some parameters of interest, like haemoglobin. For example, advanced age, which we have seen associated with an increased burden of comorbidities at baseline, may be the basis of a reduced resilience both in terms of inflammatory response and iron homeostasis. Similarly, nutritional disorders may not only negatively influence these responses, but the same individual could show lower reserves and function already at hospital admission. This was observed for patients with a phenotype of undernutrition or sarcopenia (Figure 1 and Figure 3). A few words can be spent on the trends of ferritin, neutrophil-to-lymphocyte ratio (NLR), phase angle, Barthel index, sclerostin (SOST), and dickkopf-related protein 1 (DKK1), as follows:Ferritin is a 24-unit globular protein that takes up to 4300 iron atoms to be deposited in its core. Although its serum concentration represents a small fraction of the body’s ferritin pool, low circulating levels may indicate a depletion of iron stores in the absence of infection or vitamin C deficit. This iron exhaustion may be evident at admission for patients with sarcopenic undernutrition (ferritin < 100 ng∙mL^−1^), with the highest levels being conversely seen in obese patients (Figure 1h). This additional observation is in line with our main findings of the negative influence of sarcopenia and malnutrition phenotypes on the recovery after surgery. It has been known for some time that if ferritin levels, but also haematocrit [18] and mean corpuscular volume [19], are low at hospital admission, patients will be more likely to require transfusions and encounter adverse events after major orthopaedic surgery [20]. Since ferritin is also an acute-phase protein, a postoperative increase is considered physiological as long as it does not reach excessively high levels, which was the case in patients with sarcopenia.Concerning NLR, it is considered representative of a chronic inflammatory status in the absence of trauma, and its values can be assigned to five levels: normal (<2), low (2–3.99), mild (4–5.99), moderate (6–7.99), and severe (≥8) [6]. In Figure 2f, patients with sarcopenic undernutrition are seen to be the only ones to suffer from a severe chronic inflammation at baseline. This derives from a combination of low lymphocytes (reference interval 21.8–53.1%), which has been a well-known marker of malnutrition for decades [21], and high neutrophils (reference interval 34.0–67.9%). The peak after surgery was more relevant in those who had been diagnosed with sarcopenia and undernutrition. Similarly to ferritin, these findings recall the concepts related to the acute-phase response to surgery (e.g., haematological, hormonal, metabolic, and immunologic changes) and the increased vulnerability to stress of individuals with poor body function and reserves [22]. Specifically, the surgery-derived stress is known to push the metabolism towards a negative whole-body protein balance because of an increase in protein breakdown, a concomitant release of amino acids into circulation with an impaired uptake in skeletal muscles, greater urinary nitrogen losses, a shift of protein synthesis in favour of the acute-phase reactants, and a depression of other proteins’ synthesis [23]. It is, therefore, plausible to think that the patient with sarcopenia will experience not only an altered—possibly exaggerated—immune response after major surgery but also a greater depletion of the lean mass in the postoperative period.The postoperative trends of the phase angle (Figure 3c), which is known to be directly associated with muscle mass in different age groups and health conditions [24], can help to appreciate the influence of sarcopenia on the musculoskeletal system post-surgery. It can be noted, in fact, that the non-sarcopenic patients (well nourished, pure obesity, and undernutrition) were the only ones who did not display a visible worsening of the phase angle after surgery. For what concerns the physical performance, it strongly depends on the type of aids used by the patient (e.g., crutches) as well as the possible fear of putting weight on the operated limb or the fear of falling. Other than the gait speed, we calculated the Barthel index (Figure 3h), which is a 10-item questionnaire used to evaluate the patient’s independence for what concerns feeding, bathing, grooming, dressing, reaching and using the bathroom (bowel and bladder control and toilet use), moving from bed to chair and back, ambulating on level surfaces, and climbing the stairs. Since higher scores are indicative of a greater independence, it may be recognisable for patients with pure obesity a tendency to have lower functional independence at the 10-day visit compared to their counterparts without a diagnosis. However, the difference of a few points in the index is not clinically relevant and it would remain to investigate what the recovery trends are in the long term. On the other hand, database research on 13,348 Japanese patients undergoing surgical procedures for femoral fracture found that those with a BMI over ≥27.5 kg∙m^−2^ appeared to have significantly higher functional scores at discharge than their counterparts with a lower BMI [25]. This would recall the protective effect of having a few—not too many—excess kilos in older age [6], since it can represent a useful energy reserve in the event of a trauma. However, adipose tissue is also linked to a condition of low-grade inflammation orchestrated by adipocyte-derived adipokines, which participate in the whole-body immunological crosstalk [26].The SOST and DKK1 (Figure 2g,h) were dosed in this study because they are both osteoimmunological biomarkers involved in bone remodelling [27]. Specifically, they are mediators linking the immune system to bone tissue, and inhibit the Wnt pathway, reduce osteoblast activity, and in turn promote osteoclastogenesis and bone resorption. By monitoring these two biomarkers, previous investigations in joint replacement surgery aimed at quantifying osteointegration of the endoprosthesis in order to predict early aseptic loosening [28]. Typically, for both of them, no peak should be observed in the immediate postoperative period, since an early elevation has been associated with bone nonunion [27]. Remarkably, preoperative SOST and DKK1 levels in patients with obesity were the highest among study participants. This relationship was already observed in a past cross-sectional investigation [29], further associating high levels of SOST with insulin resistance in skeletal muscles [30]. Although adipose tissue may represent a useful energy reserve, it is also true that the resulting metabolic inflammation, which we did not find in our patients based on leukocyte profiles, should not be underestimated.

### 4.1. Clinical Implications

Our findings highlight the need to include dietitians in the multidisciplinary perioperative or enhanced recovery after surgery (ERAS) team for a comprehensive nutritional assessment, diagnosis, intervention, and monitoring (i.e., Nutrition Care Process) of older adults undergoing hip arthroplasty. As nearly half of the patients were diagnosed with a nutritional disorder at admission, with sarcopenia and undernutrition emerging as the most clinically relevant predictors of greater haemoglobin decline and impaired inflammatory response, it appears to be critical to introduce the standardised NCP into preoperative risk assessment rather than relying on outdated surrogate parameters like BMI. Moreover, the observed patients’ failure to meet nutritional requirements after surgery further points to the necessity for nutritional support during recovery. These insights advocate for a shift from the hospital-centred care model, recalling the need for technological advances in the field of tele-nutrition to promote correct nutrition at the patient’s home [31], especially during rehabilitation.

### 4.2. Strengths and Limitations

Diagnosis of nutritional disorders is currently based on criteria that include body composition abnormalities, which are known to depend on the years of age, sexual dimorphism, and disease condition. Thus, statistical and predictive analyses were adjusted for age, sex, and comorbidity index to control for covariates and confounders, increasing the validity of the findings. Other efforts to address potential biases included a stratification analysis based on the nutritional disorder and blinding during statistical analyses. Concerning limitations, our cohort is not representative of the hip replacement patient population, as this was a single-centre study, and we did not include patients with emergency access for fracture nor patients with neuropsychiatric illness. The cohort was selected among patients who followed a standard protocol of surgical and anaesthetic practices, with a hospitalisation planned a priori within 3–4 days after surgery. Therefore, we could not analyse the impact on the length of hospital stay. Eligibility criteria also restricted the sample of interest to patients who were able to return for the 10-day follow-up to undergo the study evaluations, potentially losing information of patients who had a scheduled follow-up in an external clinic. Similarly, the patients willing/able to return for follow-up may have had better baseline function or social support. Moreover, the food diary has inherent limitations, including under-reporting or unknown caregiver input. Finally, although the subgroup analyses were an effort to reduce bias, the limited size of the groups being compared also had reduced power, increased uncertainty, and limited external validity. Therefore, our results should be generalisable with caution.

## 5. Conclusions

Our analytical investigation on 48 older adults undergoing primary total hip replacement found that those patients with a preoperative diagnosis of undernutrition and sarcopenia exhibited a greater haemoglobin drop and an altered immune response after surgery. Protein and energy intake inadequacies were recorded in all patients. These findings suggest that any identification of nutritional disorders should be made in advance so that there is time to apply personalised nutritional prehabilitation strategies. Postoperative initiatives to promote both proper refeeding after surgery and diet at home should also be planned. While short-term outcomes after hip arthroplasty are feasible to study, the long-term clinical and surgical consequences grounded by a nutritional disorder remain unknown.

### Future Directions

Future observational studies should recruit more subjects, include frailty assessment, and extend the monitoring of the outcomes measured up to 30 and 90 days to investigate medium- and long-term complications, rehabilitation milestones, or readmissions. Experimental studies ought to investigate what the optimal prehabilitation protocol for patients with different nutritional disorders may be, personalise the diet therapy based on the type of diagnosis, and consider an extension of nutritional support also in the postoperative period.

## Figures and Tables

**Figure 1 nutrients-17-02319-f001:**
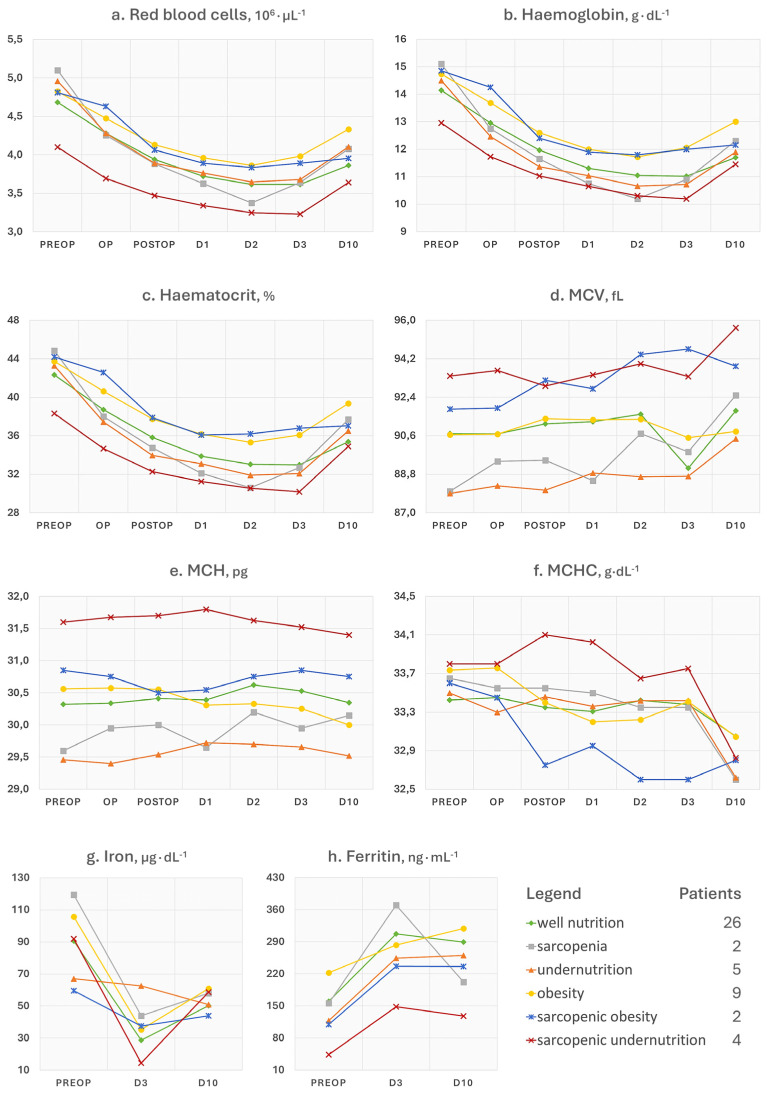
Multiple line charts of the trends of red blood cell and iron status indices stratified by nutritional disorder diagnosis. MCV = mean corpuscular volume; MCH = mean corpuscular haemoglobin; MCHC = mean corpuscular haemoglobin concentration.

**Figure 2 nutrients-17-02319-f002:**
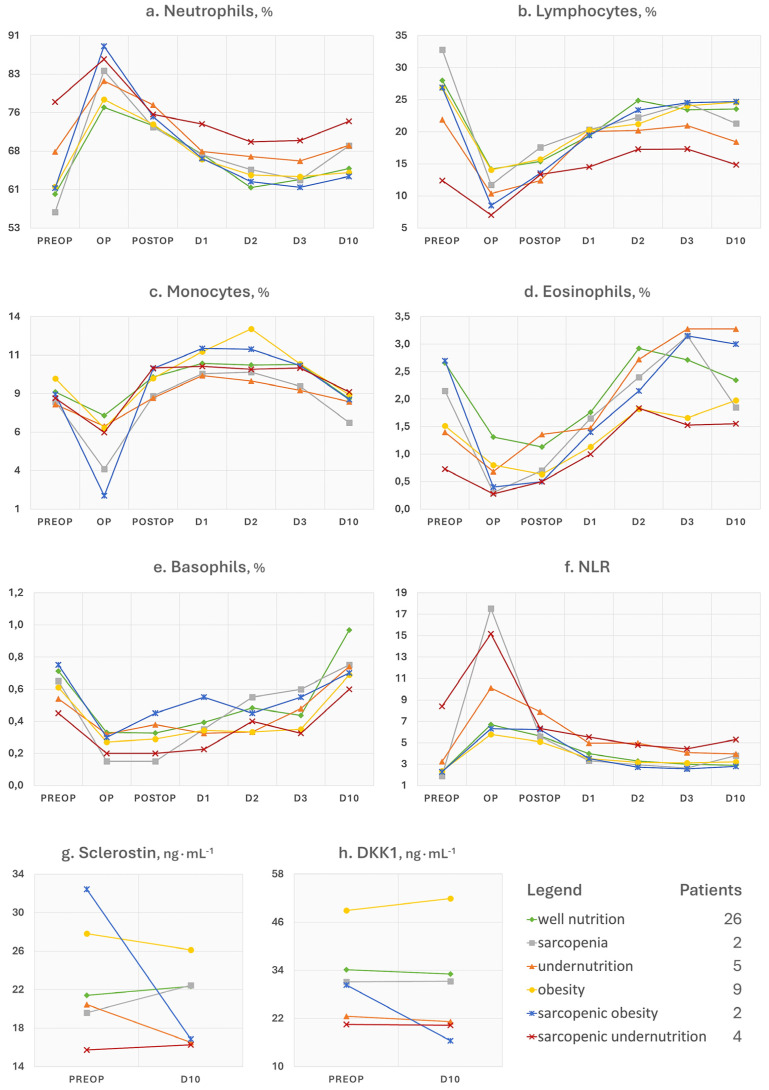
Multiple line charts of the trends of immune and inflammation markers stratified by nutritional disorder diagnosis. NLR = neutrophil-to-lymphocyte ratio; DKK1 = dickkopf-related protein 1.

**Figure 3 nutrients-17-02319-f003:**
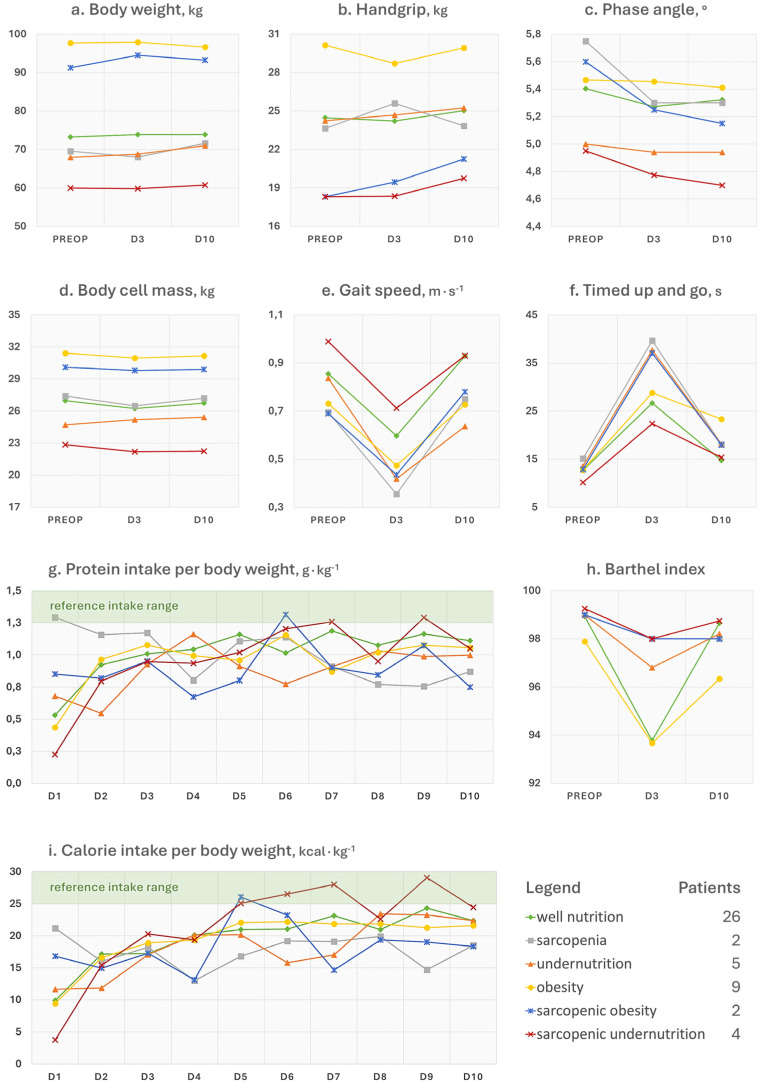
Multiple line charts of the trends of physical status and performance and protein–energy intakes stratified by nutritional disorder diagnosis. Food intakes were adjusted for ideal body weight derived from Lorentz equations of men (height, cm − 100 − [(height, cm − 150) ÷ 4]) and women (height, cm − 100 − [(height, cm − 150) ÷ 2.5]). Reference intake ranges of proteins and calories are highlighted in light green.

**Table 1 nutrients-17-02319-t001:** Preoperative characteristics of patients undergoing hip replacement surgery.

Parameter	Cohort (*n* = 48)	Females (*n* = 22)	Males (*n* = 26)
Age, years	71.54 (6.41) [61; 84]	71.73 (6.71) [63; 84]	71.38 (6.27) [61; 83]
BMI, kg∙m^−2^	27.09 (4.89) [20; 40]	25.71 (4.49) [20; 37]	28.26 (4.99) [22; 40]
CCI	3.42 (1.19) [2; 7]	3.14 (0.99) [2; 5]	3.65 (1.32) [2; 7]
LOS, days *	3.90 (0.72) [3; 5]	4.00 (0.62) [3; 5]	3.81 (0.80) [3; 5]

Data are reported as mean (SD) [min; max]. BMI = body mass index; CCI = Charlson comorbidity index; LOS = length of stay; * = including the day of admission, usually the day before the operation.

**Table 2 nutrients-17-02319-t002:** Preoperative values of red blood cell and iron status indices across nutritional disorders.

Parameter	Well Nutrition (*n* = 26)	Undernutrition (*n* = 5)	Sarcopenia (*n* = 2)	Obesity (*n* = 9)	Sarcopenic Undernutrition (*n* = 4)	Sarcopenic Obesity (*n* = 2)
RBC	4.68 (0.52)	4.96 (0.50)	5.10 (0.18)	4.82 (0.41)	4.10 (0.45)	4.81 (0.21)
Hb	14.14 (1.34)	14.50 (0.58)	15.10 (0.42)	14.73 (1.36)	12.95 (1.57)	14.85 (1.20)
Ht	42.33 (4.02)	43.28 (1.28)	44.85 (0.49)	43.74 (4.54)	38.33 (4.90)	44.20 (3.39)
MCV	90.70 (4.55)	87.90 (7.64)	88.00 (4.10)	90.66 (4.49)	93.40 (4.15)	91.85 (3.04)
MCH	30.32 (1.80)	29.46 (3.06)	29.60 (0.28)	30.57 (1.61)	31.60 (1.94)	30.85 (1.20)
MCHC	33.43 (1.12)	33.50 (1.07)	33.65 (1.34)	33.73 (1.19)	33.80 (1.18)	33.60 (0.14)
Iron	90.64 (27.36)	67.00 (14.54)	119.50 (51.62)	105.67 (23.32)	92.00 (19.78)	59.50 (23.33)
Ferritin	160.35 (123.08)	118.00 (83.39)	156.00 (94.75)	222.22 (208.49)	43.75 (14.93)	110.00 (76.37)

Data are reported as mean (SD). RBC = red blood cells, 10^6^∙μL^−1^; Hb = haemoglobin, g∙dL^−1^; Ht = haematocrit. %; MCV = mean corpuscular volume, fL; MCH = mean corpuscular haemoglobin, pg; MCHC = mean corpuscular haemoglobin concentration, g∙dL^−1^; iron = iron, μg∙dL^−1^; Ferritin = ferritin, ng∙mL^−1^.

**Table 3 nutrients-17-02319-t003:** Preoperative values of immune and inflammation parameters across nutritional disorders.

Parameter	Well Nutrition (*n* = 26)	Undernutrition (*n* = 5)	Sarcopenia (*n* = 2)	Obesity (*n* = 9)	Sarcopenic Undernutrition (*n* = 4)	Sarcopenic Obesity (*n* = 2)
Neut	59.70 (6.41)	68.08 (4.94)	56.15 (11.53)	61.18 (5.48)	77.92 (6.19)	60.90 (2.40)
Lymp	28.02 (5.52)	21.90 (4.25)	32.80 (11.31)	26.92 (4.09)	12.40 (6.25)	26.90 (0.71)
NLR	2.29 (0.91)	3.24 (0.89)	1.88 (1.00)	2.35 (0.59)	8.40 (5.96)	2.27 (0.15)
Mono	8.91 (2.00)	8.08 (1.97)	8.25 (0.49)	9.82 (3.67)	8.50 (1.39)	8.75 (1.91)
Eosi	2.66 (2.90)	1.40 (1.14)	2.15 (0.07)	1.51 (0.72)	0.72 (0.61)	2.70 (0.28)
Baso	0.71 (0.28)	0.54 (0.27)	0.65 (0.21)	0.61 (0.28)	0.45 (0.31)	0.75 (0.07)
SOST	21.42 (7.53)	20.45 (7.36)	19.62 (5.76)	27.83 (13.34)	15.74 (8.83)	32.43 (NA)
DKK1	34.15 (15.34)	22.53 (3.30)	31.16 (9.6)	48.91 (35.78)	20.53 (10.24)	30.33 (NA)

Data are reported as mean (SD). Neut = neutrophils, %; Lymp = lymphocytes, %; NLR = neutrophil-to-lymphocyte ratio; Mono = monocytes, %; Eosi = eosinophils, %; Baso = basophils, %; SOST = sclerostin, ng∙mL^−1^; DKK1 = dickkopf-related protein 1, ng∙mL^−1^.

**Table 4 nutrients-17-02319-t004:** Preoperative scores of physical status and performance across nutritional disorders.

Parameter	Well Nutrition (*n* = 26)	Undernutrition (*n* = 5)	Sarcopenia (*n* = 2)	Obesity (*n* = 9)	Sarcopenic Undernutrition (*n* = 4)	Sarcopenic Obesity (*n* = 2)
W	73.23 (11.62)	67.94 (14.43)	69.55 (7.71)	97.71 (13.47)	59.95 (8.49)	91.25 (19.45)
HGS	24.47 (10.12)	24.24 (13.29)	23.65 (0.35)	30.14 (7.36)	18.30 (5.61)	18.30 (7.21)
PhA	5.40 (0.80)	5.00 (0.58)	5.75 (0.21)	5.47 (0.77)	4.95 (0.31)	5.60 (0.85)
BCM	26.97 (6.82)	24.70 (4.91)	27.40 (0.71)	31.43 (6.71)	22.85 (5.37)	30.10 (12.02)
10MWT	0.85 (0.21)	0.84 (0.30)	0.70 (0.23)	0.73 (0.16)	0.99 (0.24)	0.69 (0.28)
TUG	12.70 (4.64)	13.55 (3.75)	15.12 (1.59)	12.79 (4.29)	10.16 (2.38)	12.85 (0.91)
BI	98.92 (1.98)	99.00 (1.00)	99.00 (1.41)	97.89 (3.48)	99.25 (1.50)	99.00 (1.41)

Data are reported as mean (SD). W = weight, kg; HGS = handgrip strength, kg; PhA = phase angle; BCM = body cell mass, kg; 10MWT = 10 m walk test or gait speed, m∙s^−1^; TUG = timed up and go, s; BI = Barthel index.

## Data Availability

Data reported in this publication is were shared, after deidentification, immediately and indefinitely as Appendix A Raw Data along with the STROBE guidelines for reporting observational cohort studies. The original contributions presented in the study are included in the article/Appendix A, further inquiries can be directed to the corresponding author.

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
