# Peer review of "Influence of Preoperative Diagnosis of Nutritional Disorders on Short-Term Outcomes After Hip Arthroplasty: A Cohort Study of Older Adults"

_nutrients, 2025, doi:10.3390/nu17142319_

Round 1
Reviewer 1 Report
Comments and Suggestions for Authors
1. This article follows the guidelines for reporting observational cohort studies (STROBE statement). - this information should be included in the methodology section, not in the purpose of the study.
2. The statistics are correct.
3. A significant part of the discussion (as much as 60%) is in fact an extended description of the results, rather than their critical interpretation or placement in a broader literary context. Paragraphs 1–3 are almost a repetition of the summary of the results – only written in a more narrative style. - This section should be completely revised. The authors declare that they use the STROBE protocol, so I recommend that the discussion be done according to this protocol.
4. The population is too small to generalise the conclusions in this way – the conclusions should be conclusions from the study, not citations from other studies.
Author Response
Reviewer 1
- This article follows the guidelines for reporting observational cohort studies (STROBE statement). - this information should be included in the methodology section, not in the purpose of the study.
RESPONSE: We moved the information in section 2.4.
- The statistics are correct.
RESPONSE: Thank you for the recognition.
- A significant part of the discussion (as much as 60%) is in fact an extended description of the results, rather than their critical interpretation or placement in a broader literary context. Paragraphs 1–3 are almost a repetition of the summary of the results – only written in a more narrative style. - This section should be completely revised. The authors declare that they use the STROBE protocol, so I recommend that the discussion be done according to this protocol.
RESPONSE: We rewrote the initial part of the discussion. Of note, the STROBE checklist highlights that the discussion should also summarise the key results with reference to the study objectives. So we kept the summary but reduced its length. Moreover, it is important to point out that the potential reader may not necessarily be able to interpret the statistics reported in the section 3. Results, so we thought of narrating the findings in the initial part of the discussion.
We moved in the section 2.4. Statistics the information regarding the focused analysis: “…No analysis was planned on the other collected parameters and recovery measures to avoid the risk of multiplicity and the likelihood of finding spurious associations. Moreover, different variables would have shown overlapping trends (e.g., haemoglobin and haematocrit), possibly conveying redundant information or adding an analytical complexity without proportionally enhancing the interpretation…”.
In the section 4. Discussion, we corrected as follows: “…The objective of this study was to explore the influence of five different nutritional disorders (undernutrition, sarcopenia, obesity, sarcopenic undernutrition, sarcopenic obesity) on haemoglobin drop, neutrophils’ surge, and gait speed recovery within 10 days after elective hip replacement surgery. These recovery measures are considered clinically meaningful in orthopaedic practice and can serve as robust proxies for the acute phase response and regain of autonomy post-surgery. A total of 22 patients, up to 45.8% of the study sample, were diagnosed with a nutritional disorder. Specifically, the sarcopenia phenotype was diagnosed in 16.7%, undernutrition in 18.8%, and obesity in 22.9%. Among these, the multivariate and elastic net analyses found that among various patient factors, nutritional disorders, particularly sarcopenia and undernutrition, were consistently the most influential on early postoperative recovery after hip arthroplasty. Specifically, patients with sarcopenia and undernutrition were those experiencing the worst 72-hour decline of haemoglobin (discharge), with the sarcopenia phenotype explaining 30.3% of haemoglobin change variability. These findings suggest that sarcopenia may cause a compromised erythropoietic response post-surgery, possibly due to underlying altered iron metabolism. Moreover, patients with co-existing sarcopenia and undernutrition had a blunted postoperative neutrophil peak, which is consistent with a basal immune dysfunction, and the undernutrition phenotype was also the most significant influencer of the magnitude of the recovery of walking autonomy at follow-up. These key results emphasise that beyond patient’s age, comorbidity burden, and phenotypes of overnutrition (i.e. obesity), a nutritional disorder with under-lying reduced body function and reserves can play a critical role in determining early physiological response after a major orthopaedic surgery. Still, compared to age that was nevertheless associated in our cohort with prolonged neutrophil elevation (i.e. poorer re-activity or a delayed resolution of the inflammatory response) and greater gait impairment at discharge, malnutrition has the advantage of being a modifiable determinant of health. Giving the novelty of diagnostic criteria for the various nutritional disorders, few studies are available. A cross-sectional survey has recently investigated the prevalence of sarcopenia and obesity in total joint replacement patients in the United Stated based on body composition analysis, confirming that prevalence of sarcopenia is around 15%, with approximately 65% of patients having obesity [11]. While sarcopenia occurrence is more in line with our results, the second finding is likely a reflection of the different prevalence of obesity in the general Italian and American populations. Another paper recently re-ported the findings from database research conducted in Spain that mainly aimed at investigating the association of undernutrition and several outcome measures after total hip replacement [15]. Compared to what we observed, the authors found a higher prevalence of both undernutrition, around 38.4%, and obesity, which was around 40%. This data can be due to the fact that our sample consisted of subjects who were on average 4 years younger, thus missing the burden of ageing on body composition. The Spanish authors also reported that undernutrition was significantly more frequent in the group of patients with postoperative complications, encompassing anaemia requiring transfusions, acute kidney injury, and surgical site infection. This is more in line with our observations, although our hospital standard of care included more stringent surgical and anaesthetic practices for the patients selected in this study, which may have excluded more complex patients for whom the complication rates may have been evident. In contrast to other studies, a diagnosis of obesity was not associated with negative effects on short-term recovery measures in our sample. This may be due to the fact that our patients with obesity had a mean BMI of 33.5 kg∙m-2, assigning them to the class 1 obesity. Studies that reported correlations between BMI-derived obesity and increased complications after joint re-placement surgeries predominantly focused on patients with class 3 obesity or higher [10,16]. Regarding postoperative dietary intakes, they were far from the optimal reference per kilo of body weight of 25-30 kcal and 1.3-1.5 grams of proteins, respectively. The failure of older orthopaedic patients to meet the nutritional needs is a long-known issue that has already been attributed to several aetiologies, including the ageing-derived early satiety, surgery-derived immobility, and polypharmacotherapy [2]. Since it is not interpretable from our results, it remains to be defined whether the different food intakes by sex, age, comorbidity burden, and nutritional diagnosis could be partly influenced by different eating assistance in the ward. Nevertheless, the presence of a nutritional disorder can make things worse, suggesting that malnourished patients may exhibit distinct eating behaviours in response to surgery. Patient education aimed at achieving postoperative dietary needs may be helpful if already planned preoperatively [17].”.
- The population is too small to generalise the conclusions in this way – the conclusions should be conclusions from the study, not citations from other studies.
RESPONSE: We removed citations from other studies and revised the section 5. Conclusions to make it more balanced as follows: “…Our analytical investigation on 48 older adults undergoing primary total hip re-placement found that those patients with a preoperative diagnosis of undernutrition and sarcopenia exhibited a greater haemoglobin drop and an altered immune response after surgery. Protein and energy intakes inadequacies were recorded in all patients. These findings suggest that any identification of nutritional disorders should be made in advance so that there is time to apply personalised nutritional prehabilitation strategies. Postoperative initiatives to promote both proper refeeding after surgery and diet at home should also be planned.”.
Reviewer 2 Report
Comments and Suggestions for Authors
This study represents an analytical investigation that aims to study different recovery trends up to ten days after hip replacement surgery in relation to the presence of a diagnosed nutritional disorder that was assessed at hospital admission. The manuscript follows the existing guidelines that assess the reporting of observational cohort studies (STROBE statement).
Please address the following comments and suggestions:
Please expand the Introduction section by assessing the published scientific literature regarding your research topic.
Conclusions section: Please use only information based on your findings as in the conclusions section there should not be any references to the scientific published literature.
Author Response
Reviewer 2
This study represents an analytical investigation that aims to study different recovery trends up to ten days after hip replacement surgery in relation to the presence of a diagnosed nutritional disorder that was assessed at hospital admission. The manuscript follows the existing guidelines that assess the reporting of observational cohort studies (STROBE statement).
Please address the following comments and suggestions:
- Please expand the Introduction section by assessing the published scientific literature regarding your research topic.
RESPONSE: we expanded the introduction with new information and references as follows “…Malnutrition has historically been identified both in orthopaedic research and clinical practice based on surrogate parameters, such as body mass index (BMI), circulating levels of some analyte of nutritional interest like albumin, transferrin, or total lymphocyte counts, and composite equations [6,7]. There is plenty of evidence associating low circulating levels of these surrogate parameters to deleterious consequences in major orthopaedic surgery, ranging from delayed recovery to higher risk of infections of the joint endo-prosthesis or surgical site [8,9]. However, these metrics come with limitations [10] and do not conform to the recognised diagnostic criteria. The most recent studies have begun to use assessment methods that are considered more appropriate for malnutrition and other nutritional disorders like sarcopenia, such as the quantities of fat and lean mass, strength, and function [11,12].”.
- Conclusions section: Please use only information based on your findings as in the conclusions section there should not be any references to the scientific published literature.
RESPONSE: We removed references to the scientific published literature and revised the section 5. Conclusions to make it more balanced as follows: “…Our analytical investigation on 48 older adults undergoing primary total hip re-placement found that those patients with a preoperative diagnosis of undernutrition and sarcopenia exhibited a greater haemoglobin drop and an altered immune response after surgery. Protein and energy intakes inadequacies were recorded in all patients. These findings suggest that any identification of nutritional disorders should be made in advance so that there is time to apply personalised nutritional prehabilitation strategies. Postoperative initiatives to promote both proper refeeding after surgery and diet at home should also be planned.”.
Reviewer 3 Report
Comments and Suggestions for Authors
Thank you for your interesting paper addressing an important issue in both pre- post-operative care.
Whilst you describe the focus of the research/analysis to be on three focus outcome measures the paper is including a huge amount of analysis of many variables using complex data analysis for what are very small patient numbers within each malnutrition group. I would suggest that if three variables have been chosen as the most meaningful then this should be evident throughout each section of the paper. It currently reads rather disjointed. Whilst that is my overall impression of the work and I feel it would be beneficially if that were addressed, there are some specific comments which should be responded to:
Dietary food records (line 138). Please state for how long and to what level of detail these were conducted. i.e. did patients weight food? compliance level?
line 135 - typo 1o should read 10
line 168-169 - The wording here seems to imply that 48 participants have a nutritional disorder but this is not the case.
Table 1. Check LOS data. The overall lOS or the cohort can't be 3.09 days of the mean for males and female patients respectively is 4 and 4.14 days.
Tables 2, 3 and 4. Please include the number of patients in each group as you have done in figure legends.
Line 321 - typo rage should read range
Lines 327 -334 - Here you reiterate from your intro that your are focusing your paper on three outcomes measures and explain why this is the case. However, this is not what you have done in your results section where you have presented 'everything'. If you address this the paper will be clearer.
Author Response
Reviewer 3
Thank you for your interesting paper addressing an important issue in both pre- post-operative care.
Whilst you describe the focus of the research/analysis to be on three focus outcome measures the paper is including a huge amount of analysis of many variables using complex data analysis for what are very small patient numbers within each malnutrition group. I would suggest that if three variables have been chosen as the most meaningful then this should be evident throughout each section of the paper. It currently reads rather disjointed. Whilst that is my overall impression of the work and I feel it would be beneficially if that were addressed, there are some specific comments which should be responded to:
- Dietary food records (line 138). Please state for how long and to what level of detail these were conducted. i.e. did patients weight food? compliance level?
RESPONSE: We integrated the text with the information required. Line 139 as follows “…Regarding patient’s diet at home, participants have completed a paper-based quantitative food diary (weighting of food required) in the first 10 days after the intervention, returning it at the final visit.”; line 306 as follows “…All 48 patients assessed for the presence of a nutritional disorder completed and re-turned the paper food diary at the follow-up visit. In Figure 3, the dietary intakes as estimated from the calculations are shown”.
- line 135 - typo 1o should read 10
RESPONSE: We corrected.
- line 168-169 - The wording here seems to imply that 48 participants have a nutritional disorder, but this is not the case.
RESPONSE: We corrected at line 177 as follows “…The presence or absence of a nutritional disorder could be assessed in 48 out of 60 participants…”.
- Table 1. Check LOS data. The overall LOS or the cohort can’t be 3.09 days of the mean for males and female patients respectively is 4 and 4.14 days.
RESPONSE: Thank you for the correction, we rectified the values indicated in Table 1.
- Tables 2, 3 and 4. Please include the number of patients in each group as you have done in figure legends.
RESPONSE: we included the number of patients in each Table 2, 3, and 4.
- Line 321 - typo rage should read range
RESPONSE: We corrected.
- Lines 327 -334 - Here you reiterate from your intro that you are focusing your paper on three outcomes measures and explain why this is the case. However, this is not what you have done in your results section where you have presented 'everything'. If you address this the paper will be clearer.
RESPONSE: thanks for the comment, which was widely discussed among the co-authors when we had to choose how to build the reporting of the article. We had stated that our endpoint variables were those three parameters (haemoglobin, neutrophils, and gait speed) and that we would have based our statistical analysis only on these. This was decided to avoid running into the “multiple comparisons problem”, that is, the fact that by running multiple tests simultaneously, the chance of obtaining at least one statistically significant result just by random chance increases. Therefore, in section 3.1, 3.2, and 3.3. we only reported statistical and ML analyses on haemoglobin, neutrophils, and gait speed. We did not present everything, just as we did not present statistical analyses that referred to parameters other than the three chosen. We only reported the other parameters collected in the study in Tables and Figures but presenting them to the reader in a solely descriptive manner -> on the one hand to not omit this information and on the other to avoid running into the multiple comparisons problem. We still believe this is the best reporting for the article. However, we rewrote the initial part of the discussion to present the main findings without reiterate the results.
In the section 4. Discussion, we corrected as follows: “…The objective of this study was to explore the influence of five different nutritional disorders (undernutrition, sarcopenia, obesity, sarcopenic undernutrition, sarcopenic obesity) on haemoglobin drop, neutrophils’ surge, and gait speed recovery within 10 days after elective hip replacement surgery. These recovery measures are considered clinically meaningful in orthopaedic practice and can serve as robust proxies for the acute phase response and regain of autonomy post-surgery. A total of 22 patients, up to 45.8% of the study sample, were diagnosed with a nutritional disorder. Specifically, the sarcopenia phenotype was diagnosed in 16.7%, undernutrition in 18.8%, and obesity in 22.9%. Among these, the multivariate and elastic net analyses found that among various patient factors, nutritional disorders, particularly sarcopenia and undernutrition, were consistently the most influential on early postoperative recovery after hip arthroplasty. Specifically, patients with sarcopenia and undernutrition were those experiencing the worst 72-hour decline of haemoglobin (discharge), with the sarcopenia phenotype explaining 30.3% of haemoglobin change variability. These findings suggest that sarcopenia may cause a compromised erythropoietic response post-surgery, possibly due to underlying altered iron metabolism. Moreover, patients with co-existing sarcopenia and undernutrition had a blunted postoperative neutrophil peak, which is consistent with a basal immune dysfunction, and the undernutrition phenotype was also the most significant influencer of the magnitude of the recovery of walking autonomy at follow-up. These key results emphasise that beyond patient’s age, comorbidity burden, and phenotypes of overnutrition (i.e. obesity), a nutritional disorder with under-lying reduced body function and reserves can play a critical role in determining early physiological response after a major orthopaedic surgery. Still, compared to age that was nevertheless associated in our cohort with prolonged neutrophil elevation (i.e. poorer re-activity or a delayed resolution of the inflammatory response) and greater gait impairment at discharge, malnutrition has the advantage of being a modifiable determinant of health.”
Round 2
Reviewer 3 Report
Comments and Suggestions for Authors
Thank you for your revisions.